# A label-free approach to detect ligand binding to cell surface proteins in real time

Verena Burtscher[†], Matej Hotka[†], Yang Li, Michael Freissmuth, Walter Sandtner*

Institute of Pharmacology and the Gaston H. Glock Research Laboratories for Exploratory Drug Development, Center of Physiology and Pharmacology, Medical University of Vienna, Vienna, Austria

**Abstract** Electrophysiological recordings allow for monitoring the operation of proteins with high temporal resolution down to the single molecule level. This technique has been exploited to track either ion flow arising from channel opening or the synchronized movement of charged residues and/or ions within the membrane electric field. Here, we describe a novel type of current by using the serotonin transporter (SERT) as a model. We examined transient currents elicited on rapid application of specific SERT inhibitors. Our analysis shows that these currents originate from ligand binding and not from a long-range conformational change. The Gouy-Chapman model predicts that adsorption of charged ligands to surface proteins must produce displacement currents and related apparent changes in membrane capacitance. Here we verified these predictions with SERT. Our observations demonstrate that ligand binding to a protein can be monitored in real time and in a label-free manner by recording the membrane capacitance.
DOI: https://doi.org/10.7554/eLife.34944.001

*For correspondence:
walter.sandtner@meduniwien.ac.at

[†]These authors contributed equally to this work

Competing interests: The authors declare that no competing interests exist.

## Introduction

Voltage-clamp recordings have been used for the past 70 years to assess currents through membranes, which are either resistive (i.e. ion flux through ion channels, *Hodgkin and Huxley, 1952*) or which arise from synchronized movement of charged residues or ions within the membrane electric field (i.e. gating currents, *Armstrong and Bezanilla, 1973*). The two hallmarks of electrophysiological recordings are high sensitivity and high temporal resolution: it is technically feasible to monitor currents through single molecules and to resolve conformational transitions, which occur on the microsecond scale. In addition, voltage-clamp recordings also allow for determining the cell membrane capacitance ($C_M$). Using this approach, it has been possible to detect changes in membrane area that result from the fusion of a single vesicle with the plasma membrane (*Neher and Marty, 1982*; *Fernandez et al., 1984*; *von Gersdorff and Matthews, 1994*). However, a change in membrane area is not the only factor that can affect capacitance. Several studies – most of them conducted in cells that overexpressed a membrane protein – demonstrated the contribution of the same to the $C_M$: the effect can be accounted for by mobile charges within the protein. These add to the total charge required to recharge the membrane upon prior voltage change. Voltage-gated channels were the first membrane proteins on which capacitance measurements were conducted: the total $C_M$ of a cell expressing these proteins was shown to comprise a voltage-independent component, which was ascribed to the membrane, and a voltage-dependent component, which was attributed to mobile charges within the voltage sensors of the channel (*Bean and Rios, 1989*). However, capacitance measurements also allow for extracting information on the conformational switch associated with binding of ions and/or substrate to transporters: it has, for instance, been possible to infer the conformational transition associated with phosphorylation of the $Na^+/K^+$-pump from the

**eLife digest** Living cells have a surrounding membrane that largely insulates the space inside from that outside. Yet signals from outside of the cell can still influence how that cell behaves. One way this can happen is via small molecules called ligands binding to proteins embedded in the membrane and triggering a cascade of reactions inside the cell. Studying this kind of protein-ligand interaction is important for many aspects of biology. However, such studies typically require that the ligand be first labeled in some way, which is time-consuming, costly and may alter how the ligand behaves. As such, there is a need for alternative ways to measure ligands binding to proteins in membranes.

The fact that the two sides of a membrane are insulated from each other means that they can store electrical charges. The ability to store charges is called capacitance. Theory predicts that it should be possible to detect a change in capacitance when a ligand binds to a membrane protein. Yet, though the theoretical basis of this hypothesis has been widely accepted, it had not been tested experimentally until now.

Burtscher et al. chose to focus on a membrane protein called the serotonin transporter because a large number of its ligands had already been characterized. Experiments with human cells that expressed this transporter confirmed that the binding of a ligand was indeed detectable as a change in membrane capacitance. Burtscher et al. also detected a brief electrical current across the membrane that is predicted to occur when the capacitance changes.

Ligand binding studies are especially important in therapeutics, as many drugs rely on blocking specific signaling pathways in diseased cells. As these capacitance recordings show precise real-time measurements, they could be used for drug screening in the future, all without the need to label the ligands.

DOI: https://doi.org/10.7554/eLife.34944.002

recorded change in capacitance (*Lu et al., 1995*). Similarly, binding of the co-substrate $Cl^-$ to the inward-facing conformation of the γ-aminobutyric acid transporter-1 (GAT1/SLC6A1) caused a very rapid reduction in the recorded capacitance, which was proposed to reflect the suppression of a charge moving reaction induced by binding of $Na^+$ to the outward -facing conformation of GAT1.

In the present study, we conducted current and capacitance measurements in HEK293 cells expressing human SERT. Like GAT1, SERT is a member of the solute carrier 6 (SCL6) family. The transport cycle of SERT is understood in considerable detail (*Schicker et al., 2012*; *Sandtner et al., 2014*), in particular the nature of substrate-induced currents, electrogenic steps associated with binding and release of co-substrate ions and decision points in the transport cycle (*Hasenhuetl et al., 2016*; *Kern et al., 2017*). In addition SERT, in contrast to GAT1, has a very rich pharmacology (*Sitte and Freissmuth, 2015*), which allows for probing the actions of inhibitors (*Hasenhuetl et al., 2015*), atypical inhibitors (*Sandtner et al., 2016*), substrates and releasers (*Sandtner et al., 2014*; *Kern et al., 2017*), and atypical substrates/partial releasers (*Bhat et al., 2017*) by electrophysiological recordings. Here we explored changes in apparent $C_M$ resulting from ligand binding to SERT. A reduction in $C_M$ was seen regardless of whether the ligand was the cognate substrate or an inhibitor of SERT, e.g. cocaine. The Gouy-Chapman model (*Gouy, 1909*; *Chapman, 1913*) predicts that charged ligand adsorption to membrane proteins must result in an apparent change of the $C_M$ and in ligand-induced displacement currents. We verified these predictions, by analysing both the ligand-induced currents and the related changes in $C_M$.

## Results

### Application of cocaine to HEK293 cells stably expressing SERT induces an inwardly directed peak current

When measured in the whole cell configuration, rapid application of 5-HT to SERT gives rise to an initial peak current, which is followed by a steady-state current (*Schicker et al., 2012*). The peak current is caused by a synchronized conformational rearrangement that carries 5-HT through the membrane (*Hasenhuetl et al., 2016*). The steady-state current on the other hand is contingent on the

progression of SERT through the transport cycle. Raising intracellular $Na^+$ impedes this cycle and eliminates the steady-state current (*Hasenhuetl et al., 2015*). This allows for studying the peak current in isolation as shown in the upper panel of *Figure 1A*, which depicts a representative trace elicited by 30 μM 5-HT in the presence of 152 mM intracellular $Na^+$.

We applied 100 μM cocaine to a cell expressing SERT to explore whether inhibitors of SERT also caused conformational rearrangements detectable by electrophysiological recordings. The recording in the lower panel of *Figure 1A* shows a cocaine-induced current, which was smaller than that elicited by 5-HT and which was absent in control cells (data not shown): In paired measurement, where cells were challenged with both 30 μM 5-HT and 100 μM cocaine (*Figure 1B*), we observed a constant amplitude ratio (5-HT: cocaine ~4:1).

The $K_D$ for cocaine binding to SERT has been estimated by radioligand binding assays, by the concentration dependence of its inhibitory action on substrate uptake and by electrophysiological means (*Hasenhuetl et al., 2015*). The affinity estimates provided by these measurements range from 100 nM to 3 μM (*Han and Gu, 2006*; *Hasenhuetl et al., 2015*). These concentrations are much lower than the 100 μM used in the above experiments. Accordingly, we measured the current in response to a wide range of cocaine concentrations (*Figure 1C*). It is evident from these recordings that an increase in cocaine concentration was associated with larger amplitudes and accelerated peak current decays. The time courses of the decays were adequately fit by a monoexponential function. In *Figure 1D*, we plotted the rate estimates by the fits as a function of the applied cocaine concentration. The rates increased linearly over the tested concentration range. At concentrations below 10 μM, cocaine failed to evoke detectable currents.

Binding of competitive inhibitors to SERT is dependent on $Na^+$ (*Korkhov et al., 2006*). Therefore, we measured the cocaine peak in the absence of extracellular $Na^+$: at this condition, 100 μM cocaine failed to elicit a current (*Figure 1E*). In contrast, when $Cl^-$ was removed from the extracellular solution, the cocaine-induced currents were present, but featured smaller amplitudes (*Figure 1F,G*).

## The amplitude of the cocaine-induced current depends on voltage and is larger at positive potentials

We measured the voltage dependence of the cocaine-induced current by relying on the protocol depicted in the left panel of *Figure 2A*: the membrane was clamped for 1 min to a constant voltage ranging from −50 to +30 mV. During each successive voltage-clamp, 100 μM cocaine was applied for 5 s and subsequently removed to generate a family of currents (right-hand panel of *Figure 2A*). Excessive noise precluded analysis above and below this voltage range. It is evident from *Figure 2A* that the cocaine peak became larger at positive potentials. The current-voltage relation was examined by averaging nine independent experiments after normalizing the current amplitudes to the current recorded at +30 mV: this current-voltage relation was linear over the explored voltage range and had a negative slope (*Figure 2B*).

## The absence of $Cl^-$ prevents conformational change in SERT and renders the 5-HT-induced peak current similar to the current induced by cocaine

It has long been known that extracellular $Cl^-$ is required for SERT-mediated substrate uptake (*Lingjaerde, 1971*; *Nelson and Rudnick, 1982*). However, the presence of $Cl^-$ is not required for 5-HT binding to SERT (*Nelson and Rudnick, 1982*). In fact, the absence of $Cl^-$ precludes the conversion of the substrate-loaded outward-facing transporter to the inward-facing conformation. Thus, in the following experiments, we removed $Cl^-$ from the extracellular solution to separate the event of 5-HT binding from the subsequent conformational change.

Removal of $Cl^-$ reduced the peak amplitude and the area under the peak (see *Figure 3A* for representative current traces and *Figure 3B* for the comparison of 20 independent experiments with paired recordings done in the presence and absence of $Cl^-$). We also determined the current-voltage relation in the presence (*Figure 3C,D*) and absence of $Cl^-$ (*Figure 3E,F*): in contrast to the positive slope seen in the presence of $Cl^-$ (*Figure 3C,D*), the slope was negative in the absence of $Cl^-$ (*Figure 3E,F*). These observations indicated that the absence of $Cl^-$ unmasked an event preceding the substrate-induced conformational change: this initial event had a small current amplitude and a negative slope in the current-voltage relationship.

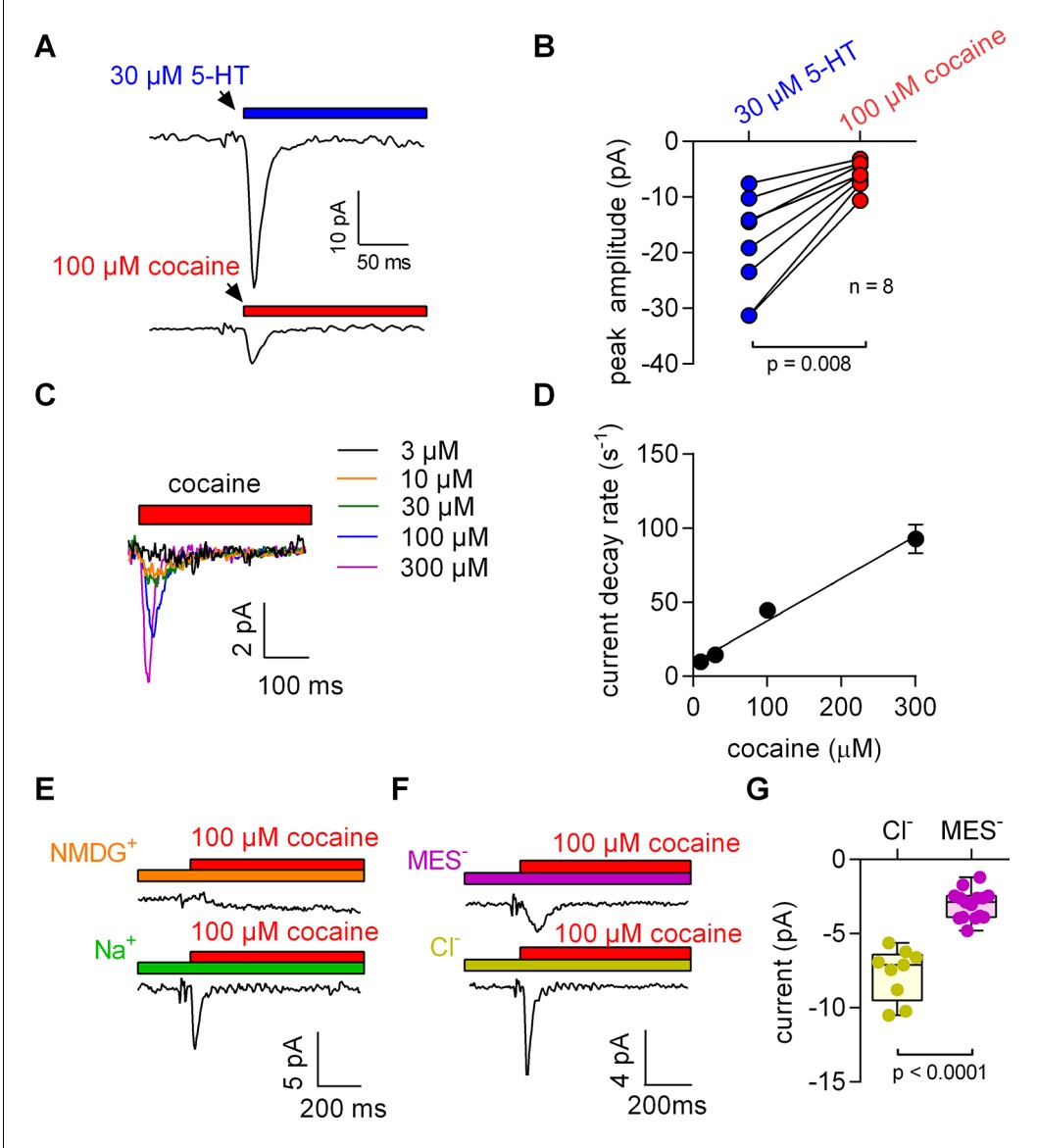

**Figure 1.** 5-HT and cocaine give rise to inwardly directed peak currents when applied to HEK293 cells stably expressing SERT. The amplitude and time constant of the decay of the cocaine-induced current are concentration dependent. (**A**) Representative currents from the same cell evoked by rapid application of 30 μM 5-HT (upper panel) or 100 μM cocaine (lower panel), respectively. The currents were recorded at 0 mV and in the presence of 152 mM intracellular $Na^+$. (**B**) Comparison of the amplitudes of the 5-HT (blue circles) and the cocaine-induced peak current (red circles). The lines connect measurements from the same cell: 5-HT: −18.8 ± 9.1 pA; cocaine: −6.0 ± 2.4 pA; n = 8; p=0.008, Wilcoxon test. (**C**) Representative currents elicited by 3 μM (black), 10 μM (orange), 30 μM (green), 100 μM (blue) and 300 μM (magenta) cocaine, respectively. The depicted currents were recorded from the same cell. (**D**) Rate of current decay as a function of cocaine concentration. The line is a linear fit to the data points (slope: $2.9*10^5 ± 0.3*10^5$ $Mol^{-1}s^{-1}$; n = 7). Data are mean ± SD. (**E**) In the absence of $Na^+$, 100 μM cocaine failed to induce a peak current. Shown are representative traces in the presence of 150 mM extracellular $NMDG^+$ (upper panel) and 150 mM extracellular $Na^+$ (lower panel), respectively. The depicted traces were recorded from the same cell. The absence of the cocaine-induced current in the presence of $NMDG^+$ was confirmed in seven independent experiments. (**F**) Representative currents in the presence of 150 mM extracellular $MES^-$ (upper panel) and 150 mM extracellular $Cl^-$ (lower panel), respectively. (**G**) The amplitude of the cocaine-induced current peak was −7.7 ± 1.7 pA (n = 9) and −3.1 ± 1.0 pA (n = 15) in the presence and absence of $Cl^-$, respectively (p<0.0001; Mann-Whitney U-test). Source files are available in *Figure 1—source data 1*.

DOI: https://doi.org/10.7554/eLife.34944.003

*Figure 1 continued on next page*

*Figure 1 continued*

The following source data is available for figure 1:

**Source data 1.** Cocaine- or 5-HT-induced transient currents recorded from SERT-expressing cells for the panels indicated.

DOI: https://doi.org/10.7554/eLife.34944.004

## The Gouy-Chapman model predicts displacement currents upon ligand binding to membrane proteins

In contrast to 5-HT, which is the cognate substrate of SERT, cocaine is not translocated. Under our recording conditions, both compounds give rise to a transient current, which may arise by two alternative mechanisms: it is conceivable that binding of cocaine or of 5-HT to SERT occurs via an induced fit, that is local rearrangements are required within the vicinity of the binding pocket to accommodate the compound. These rearrangements can be expected to force charged residues of SERT to change their position within the membrane electric field. However, it is difficult to reconcile this with the negative slope in the current-voltage relation of both the cocaine- and the 5-HT-induced peak current in the absence of $Cl^-$. The negative slope in the current-voltage relation was inconsistent with the conjecture that the cocaine-induced peak current originated from charges moving in the membrane electric field. The rules of electrostatics dictate that inwardly directed currents must increase at negative voltages, if they are carried by charges moving in response to a conformational change. This is also evident from the current-voltage relation for the 5-HT-induced peak current in the presence of chloride (*Figure 3C,D*). This current reflects the conformational rearrangement that carries substrate and co-substrates through the membrane (*Hasenhuetl et al., 2016*). Accordingly, the corresponding current-voltage relation had a positive slope (*Figure 3D*). Alternative scenarios, which produce a negative slope, include a voltage-dependent gating step or a blockage of transport at negative potentials. These possibilities, however, can be discarded because cocaine is not transported, and thus there is no gating.

Therefore, an alternative explanation is more plausible, if it can relate these currents to ligand binding without the need to invoke conformational changes: the Gouy-Chapman model provides a mathematical description of a simplified biological membrane (see scheme in *Figure 4A*) and allows for calculating the potentials at the inner and outer surface of the membrane. These potentials depend on the voltage difference between the intra- and extracellular bulk solution, the ionic composition of these solutions and the number of charges at the inner and outer membrane surfaces. In biological membranes, the charges at the surfaces comprise the phospholipid head groups and the solvent-accessible acidic or basic residues of membrane proteins. The number of charges at the

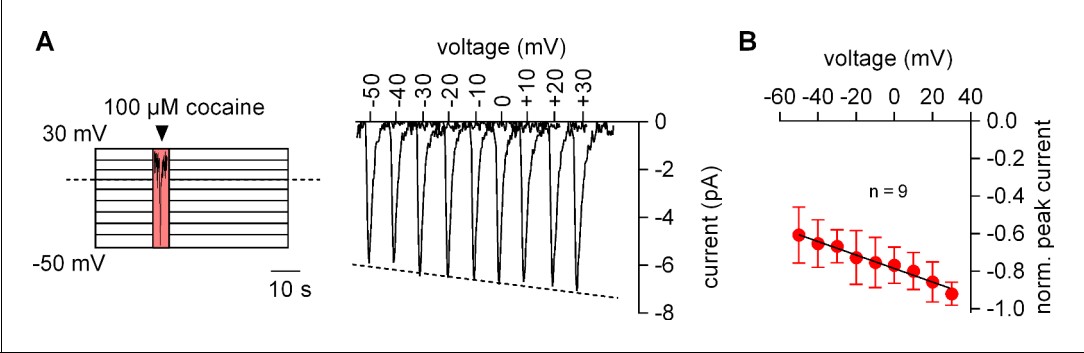

**Figure 2.** The amplitude of the cocaine peak increases at positive potentials. (**A**) The scheme illustrates the protocol employed to measure the voltage dependence of the cocaine induced peak current: cells were clamped to voltages between $-50$ and $+30$ mV for 60 s, respectively. At each potential 100 μM cocaine was applied for 5 s and subsequently removed. Shown are representative currents elicited by the protocol. (**B**) Peak currents induced by 100 μM cocaine were normalized to the largest current (n = 9). Data are mean ±SD. The black line is a linear fit to the data points (slope = $-3.5*10^{-3} \pm 5.3*10^{-4}$/mV).

DOI: https://doi.org/10.7554/eLife.34944.005

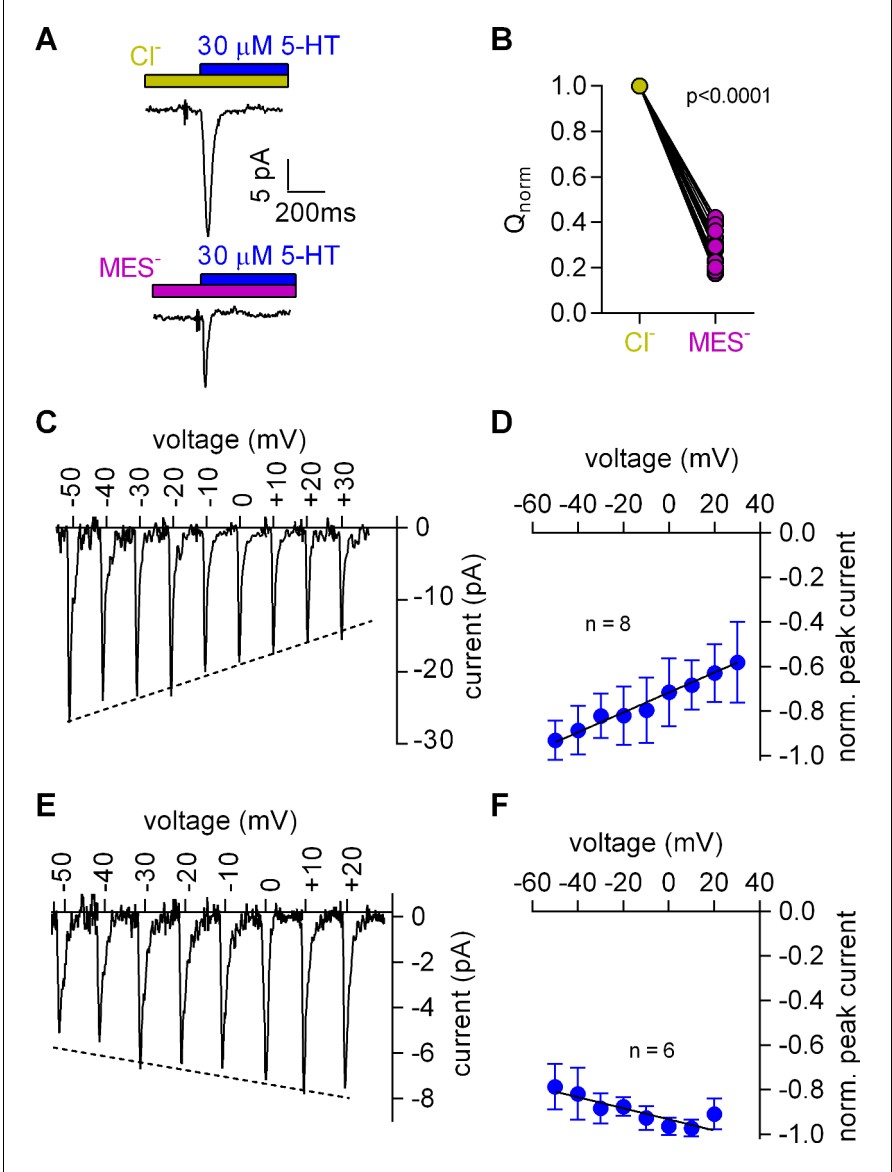

**Figure 3.** Cl⁻ removal diminishes the amplitude of the 5-HT-induced current and reverts the slope of the voltage dependence of the peak current. (**A**) Representative traces of currents evoked by 30 μM 5-HT in the presence of 150 mM extracellular Cl⁻ (upper panel) or 150 mM MES⁻ (lower panel) from the same cell. The cell was held at 0 mV. (**B**) Comparison of the normalized charge (Q) carried by the 5-HT-induced current in the presence and absence of Cl⁻. Connecting lines indicate measurements from the same cell. After removal of Cl⁻ the remaining fraction of charge was $0.28 \pm 0.075$ (n = 20). The charge before and after Cl⁻ removal was significantly different (p=0.0001; Wilcoxon test). (**C**) Representative peak currents induced by 30 μM 5-HT recorded at potentials ranging from −50 to +30 mV in the presence of 150 mM extracellular Cl⁻. (**D**) The amplitude of the 5-HT-induced peak current measured in the presence of Cl⁻ as a function of voltage. The peak current amplitudes evoked by 30 μM 5-HT were normalized to the largest current (n = 8). Data are mean ±SD. The line is a linear fit to the data points (slope = $4.5*10^{-3} \pm 5.4*10^{-4}$/mV). (**E**) Representative peak currents evoked by 30 μM 5-HT in the absence of Cl⁻ recorded at potentials ranging from −50 to +20 mV. (**F**) Normalized 5-HT-induced peak currents recorded in the absence of Cl⁻. The peak current amplitudes were normalized to the largest current (n = 6). Data are mean ± SD. The line is a fit to the data points (slope = $-2.5*10^{-3} \pm 4.9*10^{-4}$/mV).
DOI: https://doi.org/10.7554/eLife.34944.006

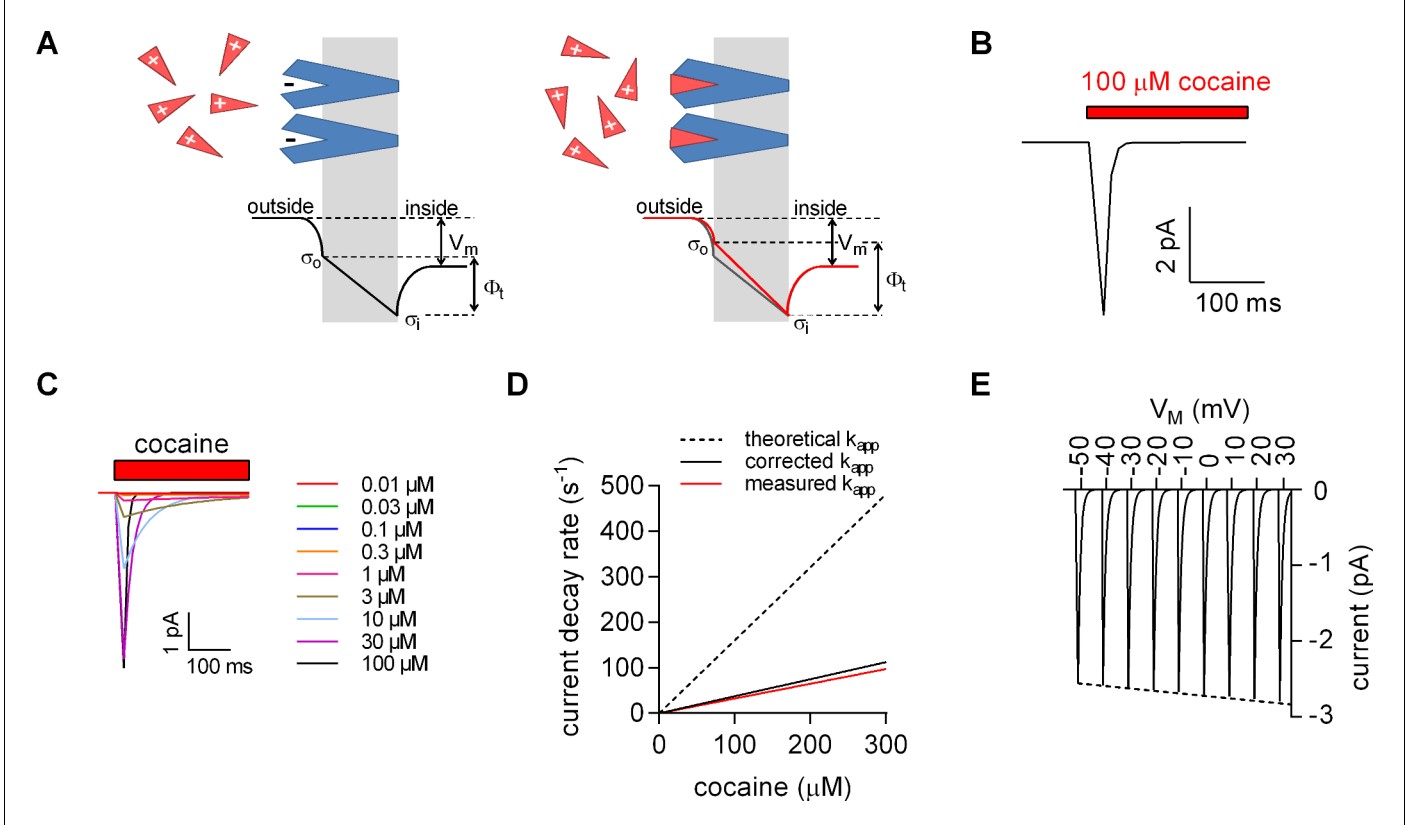

**Figure 4.** Model for ligand-induced surface charge elimination. (**A**) Scheme of a membrane (grey slab) with embedded transporters (in blue) and ligands (red triangles). The ligand is positively charged and the binding site on the protein comprises a negative charge (left panel). Upon binding, the ligand neutralizes the charge on the protein (right panel). The lines in the left (in black) and right panels (in red) indicate voltage profiles across the membrane. Binding of the ligand to the negative surface charge on the protein renders the outer surface charge potential more positive. This produces a change in transmembrane voltage ($\Delta\Phi_t$). (**B**) Displacement current predicted by the Gouy-Chapman model. The instantaneous current is calculated as: $i(t)=C_M \cdot dv(t) \cdot d(t)^{-1}$, where $C_M$ is the capacitance of the membrane and $v(t)= \Phi_t \cdot (1-e^{-k_{app}*t})$; $k_{app}$ is the apparent rate of cocaine association ($k_{app}=k_{on} \cdot$ [cocaine]$+k_{off}$). Shown is a simulated current evoked by application of 100 μM cocaine. (**C**) Simulated currents at the indicated cocaine concentrations. (**D**) Predicted rates of the current decays of the cocaine peaks as a function of the cocaine concentration (dashed line). The solid red line in the plot indicates measured rates from Figure 2B. The black solid line indicates the corrected rates (see Materials and methods, section 'Modeled and measured apparent association rates ($k_{app}$) of cocaine'). (**E**) Simulated voltage dependence of the cocaine peak. The current-voltage relation has a negative slope (slope= $-1.1*10^{-3} \pm 1.4*10^{-5}$/mV).

DOI: https://doi.org/10.7554/eLife.34944.007

The following figure supplement is available for figure 4:

**Figure supplement 1.** Schematic representation of the voltage across the membrane as predicted from the Gouy-Chapman model.

DOI: https://doi.org/10.7554/eLife.34944.008

surfaces is an important determinant of the transmembrane potential ($\Phi_t$), which represents the difference between the potentials at the opposite surfaces.

We relied on the Gouy-Chapman model to test the hypothesis illustrated in the scheme shown in *Figure 4A* (see also *Figure 4—figure supplement 1*) using the following starting parameters: (i) in our stable cell line SERT is expressed at high levels; the number of SERT molecules was estimated in saturation binding experiments of [³H]imipramine and amounted to $2 \pm 0.4 \times 10^7$ transporters/cell. Ligands of SERT including 5-HT and cocaine are positively charged. If all SERT moieties are saturated by stoichiometric binding (one positively charged ligand/SERT), the charge density at the outer surface is expected to change by about $\Delta+0.0011$ C/m². (ii) The total surface charge density on the outer surface of untransfected HEK293 cells is about $-0.005$ C/m² (*Zhang et al., 2001*). The ligand-induced change in outer surface charge density changes $\Phi_t$ by about $-10$ mV (*Figure 4—figure supplement 1*). Because the cell membrane represents a capacitor, this must result in a displacement current. *Figure 4B* depicts simulated displacement currents based on the voltage estimates

provided by the model. The simulated currents are inwardly directed with amplitudes in the low picoampere range and thus match the observed currents.

In *Figure 4C*, we modeled the concentration dependence of the cocaine-induced peak: it is implicit to our hypothesis that the time course of ligand-induced change in $\Phi_t$ must coincide with the time course of cocaine binding. We recently determined the association rate ($k_{on}$) and dissociation rate ($k_{off}$) of cocaine for SERT by an electrophysiological approach (*Hasenhuetl et al., 2015*). These rates were used to calculate the apparent association rate ($k_{app}$, dashed line in *Figure 4D*). At low cocaine concentrations, the simulated currents compared favorably with the observed. However, at higher cocaine concentrations the predictions deviated from the measured $k_{app}$ (red solid line in Figure 4D). We attribute this discrepancy to the fact that, at concentrations exceeding 30 µM, the solution exchange by our application device ($\sim20$ s$^{-1}$) becomes rate-limiting; for technical reasons, the diffusion-limited association rate for cocaine is therefore currently inaccessible to an experimental determination. We applied a correction for the finite solution exchange rate (see Material and methods). The corrected $k_{app}$ is displayed in *Figure 4D* (black solid line).

We also calculated the current-voltage relation for the displacement current (*Figure 4E*): consistent with our observations, the synthetic data predict larger currents at positive potentials. The hypothesis that charged ligand binding results in the generation of a displacement current therefore provides a parsimonious explanation for the negative slope of the observed current-voltage relation.

## Application of cocaine to HEK293 cells expressing SERT decreases the apparent membrane capacitance

The Gouy-Chapman model can be used to calculate the change in apparent $C_M$ resulting from ligand adsorption to the extracellular surface (*Figure 5A*, see also *Figure 5—figure supplement 1*). This prediction was verified. *Figure 5B* shows a representative recording of the membrane capacitance ($C_M$) with the two other circuit parameters $R_M$ and $R_S$ upon application and subsequent removal of 100 µM cocaine to HEK293 cells expressing SERT. It is evident from this recording that there was no cross talk between circuit parameters. This effect of cocaine on $C_M$ was absent in control cells (*Figure 5C*). The reduction in $C_M$ by cocaine amounted to approx. 500 fF, which is in good agreement with the prediction (*Figure 5D*). *Figure 5E* shows the concentration dependence of the cocaine-induced decrease in apparent $C_M$. These data were adequately fit by a saturation hyperbola, which provided an estimate for the affinity of cocaine to SERT (EC$_{50}$ = 156 ± 41 nM; *Figure 5F*). This estimate is in line with the published $K_D$ of cocaine (*Hasenhuetl et al., 2015*).

Additionally, we explored the effect of 100 µM 5-HT on the apparent $C_M$ in the presence of external Cl$^-$. Application of 5-HT reduced the membrane capacitance in cells expressing SERT but not in control cells (*Figure 5G*). Upon removal of 5-HT, the membrane capacitance relaxed to the initial level in cells expressing SERT (upper panel in *Figure 5G*). In the presence of extracellular Cl$^-$ the reduction in $C_M$ induced by 100 µM 5-HT amounted to $-340 \pm 140$ fF (*Figure 5H*). We also applied increasing concentrations of 5-HT to determine the concentration-response curve for the 5-HT-induced apparent reduction in $C_M$. The resulting saturation hyperbola provided an estimate for the apparent affinity of 5-HT to SERT (EC$_{50}$ = 1.4 ± 0.1 µM, *Figure 5I*). Comparison of the apparent reduction of capacitance elicited by 100 µM cocaine and 100 µM 5-HT revealed no difference as is seen by the similar magnitude of the reduction in paired recordings (*Figure 5J*). This equivalent capacitance change was expected because 100 µM is a saturating concentration for both ligands. *Figure 5K* shows the reduction in apparent $C_M$ at a saturating concentration of 5-HT (100 µM) in the absence of extracellular Cl$^-$. As can be seen the 5-HT-induced reduction in capacitance under this condition was similar to the one observed in the presence of Cl$^-$ (cf. *Figure 5I and H*).

## The voltage dependence of the cocaine-induced capacitance change is predicted by the Gouy-Chapman model

The voltage dependence of the cocaine-induced capacitance change was analyzed using the protocol outlined in *Figure 6A*. We first measured the $C_M$ in the absence of cocaine at $-50$ mV, 0 mV and $+50$ mV and applied 100 µM cocaine thereafter. Consistent with the results shown in *Figure 5A*, application of cocaine reduced the $C_M$ in SERT-expressing cells (left panel in *Figure 6A*) but not in control cells (right panel in *Figure 6A*). We also simulated the recordings in SERT-expressing cells using the Gouy-Chapman model: these synthetic data recapitulated the capacitance recordings (cf.

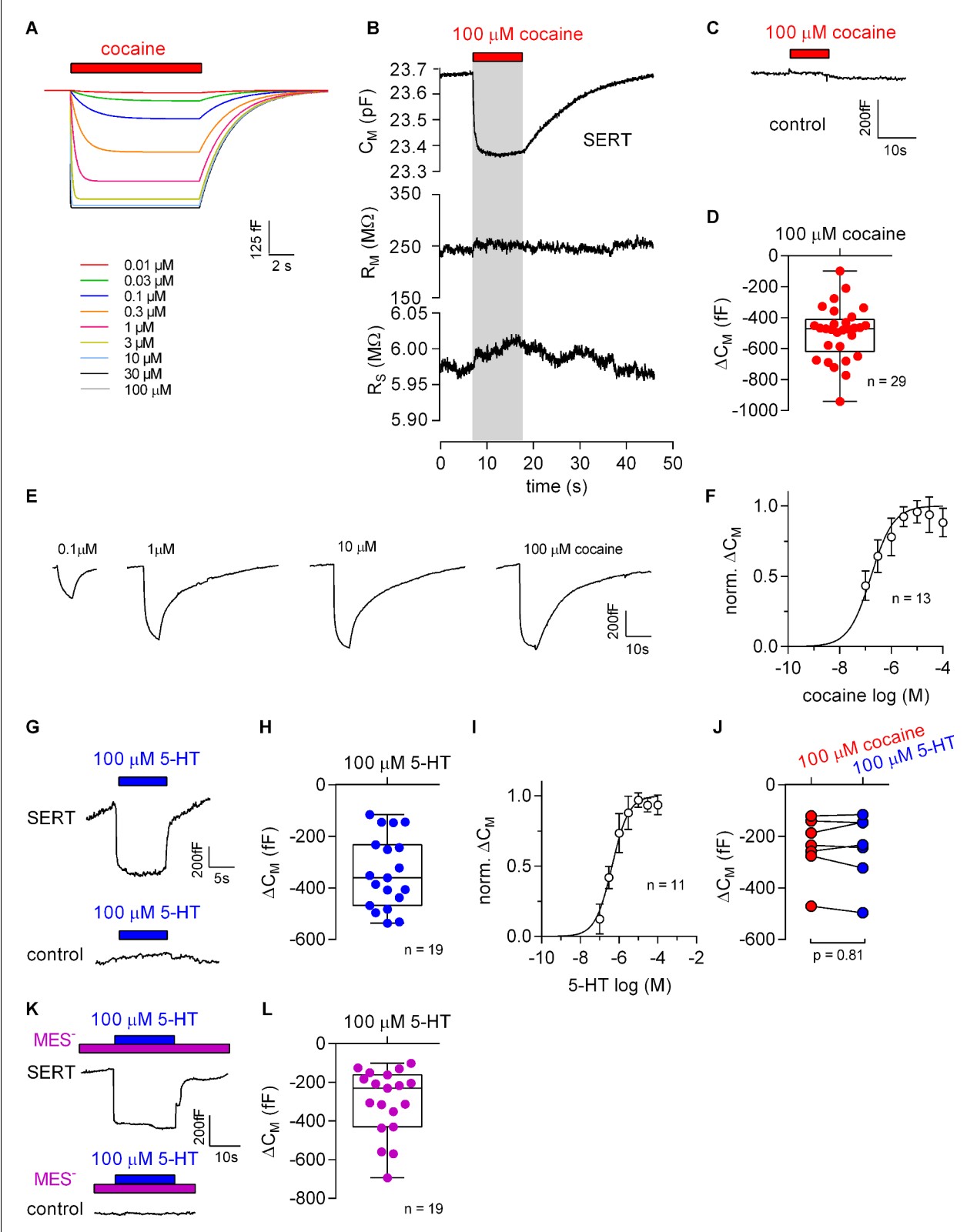

**Figure 5.** Cocaine and 5-HT binding to SERT results in a reduction of apparent membrane capacitance. (A) Predicted change in $C_M$ by binding of cocaine by the Gouy-Chapman model. The traces are the simulated response to the indicated cocaine concentrations. (B) Representative change in capacitance recorded in the presence of 100 μM cocaine in a SERT-expressing cell. Also shown are the other circuit parameters, $R_M$ and $R_S$ derived from the same recording. (C) Cocaine did not change $C_M$ in control cells. (D) Plot of the change ($\Delta C_M$) induced by 100 μM cocaine (n = 29; $\Delta C_M$ = 

*Figure 5 continued on next page*

*Figure 5 continued*

−495 ± 175 fF). (**E**) Representative traces of the cocaine-induced apparent reduction in capacitance at the indicated concentrations. The recordings are from the same cell. (**F**) Concentration-response curve for the cocaine-induced change in membrane capacitance (n = 13), which was normalized to the maximal cocaine-induced $\Delta C_M$. Data are mean ± SD. The solid line was generated by fitting the data to a rectangular hyperbola yielding an $EC_{50}$ = 164 ± 41 nM. (**G**) Representative change in capacitance elicited by 100 μM 5-HT in a SERT-expressing cell and in a control cell. (**H**) Plot of the change ($\Delta C_M$) induced by 100 μM 5-HT (n = 19, $\Delta C_M$ = −340 ± 140 fF). (**I**) Concentration-response curve for the 5-HT-induced change in $C_M$ (n = 11), which was normalized to the largest change in capacitance. Data are mean ± SD. The solid line was generated by fitting the data to a rectangular hyperbola yielding an $EC_{50}$ = 0.4 ± 0.1 μM. (**J**) Comparison of the response to 100 μM cocaine and 100 μM 5-HT, respectively (n = 8). The lines connect measured values from the same cell. The data are not significantly different (p=0.81; Wilcoxon test). (**K**) Representative change in capacitance elicited by 100 μM 5-HT in the absence of extracellular $Cl^-$ in SERT-expressing cells and control cells. (**L**) Plot of the change ($\Delta C_M$) induced by 100 μM 5-HT in the absence of extracellular $Cl^-$ (n = 19, $\Delta C_M$ = −299 ± 170 fF). Two-group comparison of 5-HT-induced $\Delta C_M$ in the presence and absence of extracellular Cl- revealed no significant difference (p=0.27, Mann-Whitney U-test). Source files are available in *Figure 5—source data 1*.
DOI: https://doi.org/10.7554/eLife.34944.009

The following source data and figure supplements are available for figure 5:

**Source data 1.** Cocaine- or 5-HT-induced apparent changes in capacitance recorded from SERT-expressing cells for the panels indicated.
DOI: https://doi.org/10.7554/eLife.34944.012
**Figure supplement 1.** Modeling of the ligand-induced change in membrane capacitance.
DOI: https://doi.org/10.7554/eLife.34944.010
**Figure supplement 2.** Mobile charges of SERT do not affect the measurements of membrane capacitance.
DOI: https://doi.org/10.7554/eLife.34944.011

left panel in *Figure 6A* and *Figure 6B*). It is important to note that the Gouy-Chapman model also predicted the voltage-dependent change in capacitance observed in control cells (*Figure 6—figure supplement 1*). The capacitance recordings from eight independent experiments are summarized in *Figure 6C*. The black and the grey dashed lines represent the linear regression through the actual recordings and the changes predicted the Gouy-Chapman model, respectively. It is evident that (i) the cocaine-induced decrease in capacitance depends on the voltage in a linear manner, and (ii) that the experimental observations and the synthetic data are in reasonable agreement. However, there is one notable difference: in the absence of cocaine, changes in voltage were followed by a slow change in capacitance in SERT-expressing cells, in particular when the prior voltage had been +50 mV (inset *Figure 6A*). The model does not account for this feature. Nevertheless, this slow change in capacitance is specific to SERT-expressing cells, because it is absent in control cells and inhibited by cocaine. We surmise that the transient change in capacitance reflects entry into/return from a state of SERT, which is occupied at very positive potentials.

## The cocaine-induced current peak carried by SERT exemplifies a more general phenomenon

In the Gouy-Chapman model no reference is made to specific proteins or ligands. In this context, SERT is treated as an entity, which provides a ligand binding site and cocaine is a charged agent that adsorbs to it. This description is general. Accordingly, the model can make two predictions: (i) similar currents are expected to arise upon application of any charged ligand to cells expressing cognate membrane protein. (ii) Similarly, inhibitors other than cocaine are expected to induce transient currents when applied to cells expressing SERT. We tested both predictions.

First, we challenged HEK293 cells expressing the dopamine transporter with cocaine: this gave rise to currents, which had a current-voltage relation with a negative slope (*Figure 7—figure supplement 1*). To test the second prediction we used desipramine binding to SERT. Consistent with the results obtained with cocaine, we detected an inwardly directed peak current upon rapid application of 100 μM desipramine (*Figure 7A*). We also recorded a drop in apparent $C_M$ when cells expressing SERT were challenged with 10 μM desipramine (representative trace in the left panel of *Figure 7B*). This was not seen in control cells (right panel of *Figure 7B*). At concentrations exceeding 10 μM, desipramine increased $C_M$ in both SERT-expressing and control cells (representative traces in *Figure 7C*). Thus, in SERT-expressing cells, the concentration-response curve for the desipramine-induced change in capacitance was biphasic with a decline in the low concentration range followed by an increase in the high concentration range (full symbols in *Figure 7D*). In contrast, only the

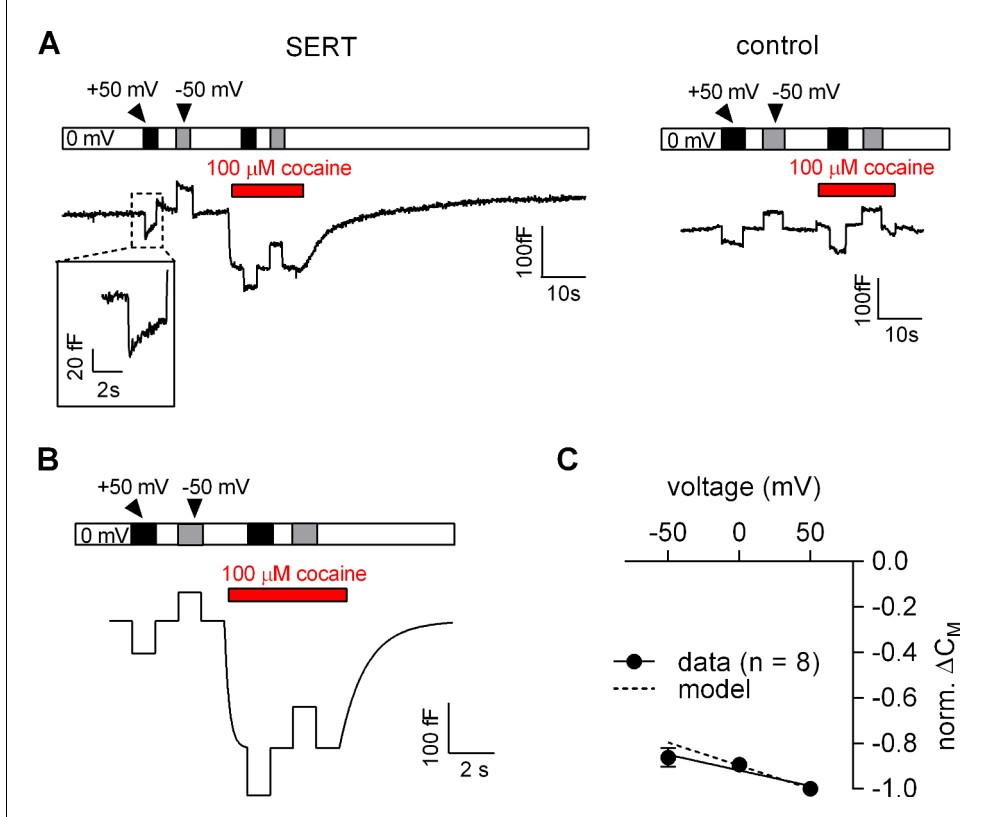

**Figure 6.** Voltage dependence of the reduction in apparent membrane capacitance by cocaine. (**A**) The capacitance was recorded in a SERT-expressing cell (left panel) and in a control cell (right panel). The holding potential was changed to +50 or −50 mV as shown by the bar in the absence and presence of 100 µM cocaine. (**B**) The experiment in the left panel in *Figure 6A* was simulated by a model based on the Gouy-Chapman equation. (**C**) The capacitance was recorded as outlined in *Figure 6B* (-50 mV, 0 mV and 50 mV) and the cocaine-induced change was normalized by setting the amplitude of the capacitance change at +50 mV to −1. Data are means ± SD (n = 6). The solid line was drawn by linear regressions (slope = $−1.4*10^{-3} ± 1.7*10^{-4}$/mV). The dashed line indicates the voltage dependence of the cocaine-induced capacitance change predicted by the Gouy-Chapman model.

DOI: https://doi.org/10.7554/eLife.34944.013

The following figure supplement is available for figure 6:

**Figure supplement 1.** Voltage dependence of the apparent membrane capacitance as predicted by the Gouy-Chapman model.

DOI: https://doi.org/10.7554/eLife.34944.014

ascending limb of the concentration-response curve was seen in untransfected control cells (empty symbols in *Figure 7D*).

## Desipramine binding to the cytosolic surface increases the membrane capacitance

Desipramine is an amphiphilic cation. The increase in capacitance, which was seen at desipramine concentrations ≥ 30 µM in both SERT-expressing and untransfected HEK293 cells, can be rationalized by assuming that desipramine permeated the cell membrane and adsorbed to the inner surface. We carried out a simulation based on this conjecture, which recapitulated the capacitance recordings (*Figure 7E*). Thus, the rise in capacitance was the expected consequence of desipramine adsorption to the inner surface of the membrane. We verified this conclusion by lowering the pH in the bath solution to 5.5. This manipulation increases the fraction of protonated species of desipramine at the expense of the diffusible uncharged species (pKa = 10.2). Lowering the pH is therefore predicted to diminish the SERT-independent increase in $C_M$, if this action is contingent on the

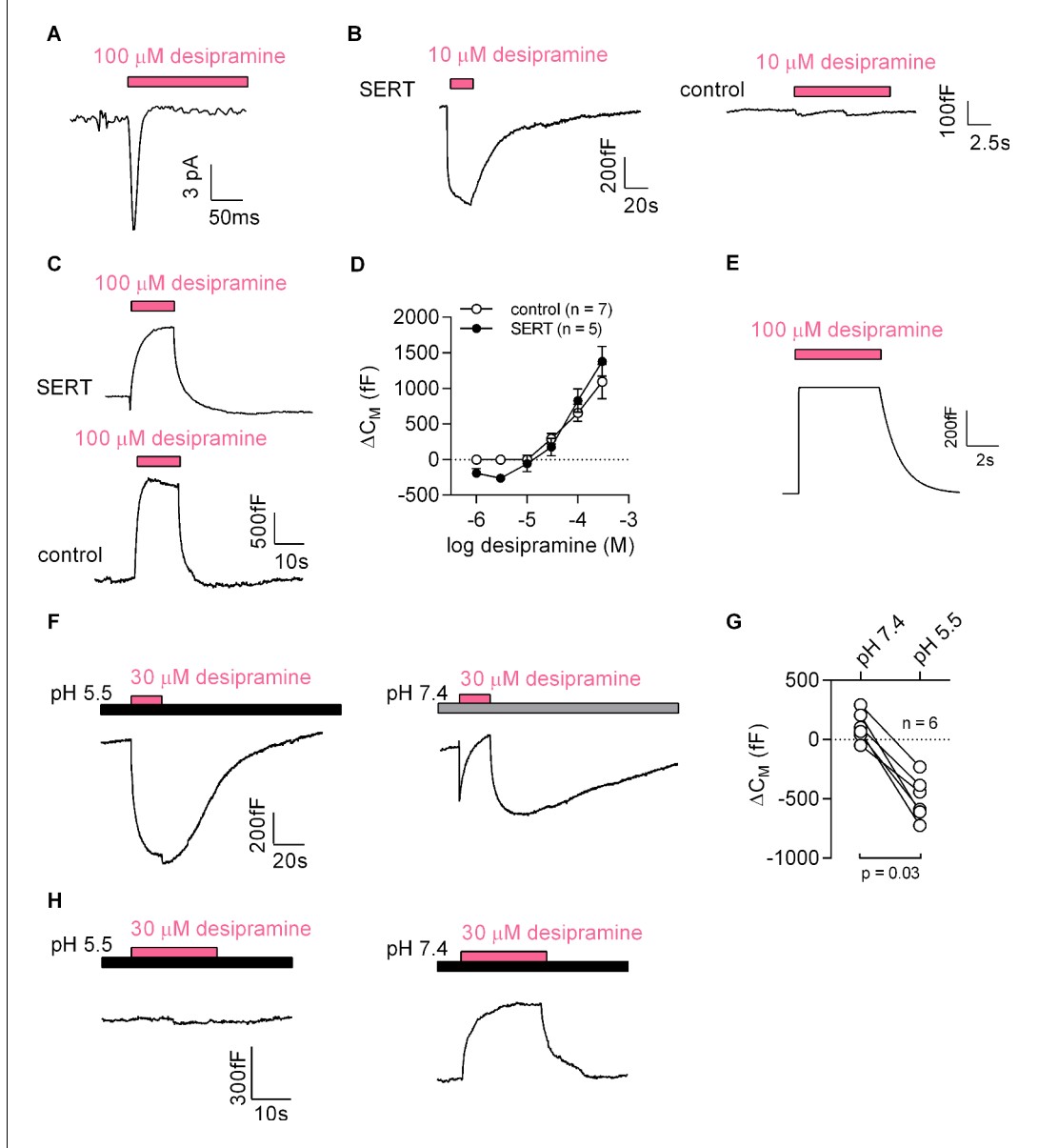

**Figure 7.** Desipramine binds to SERT at low concentrations and changes the surface charge density at the inner leaflet of the membrane at high concentrations. (A) Representative displacement current evoked by the application of 100 µM desipramine to a SERT-expressing cell. (B) Application of 10 µM desipramine resulted in a reduction of $C_M$ in a cell overexpressing SERT (left panel) but not in control cells (right panel). (C) Application of 100 µM desipramine gave rise to an apparent increase in $C_M$ in both SERT-expressing cells and control cells. (D) Concentration-dependence of the desipramine-induced change in capacitance ($\Delta C_M$) in SERT-expressing cells and in control cells. Data are means ±SD (control: n = 7, SERT: n = 5). (E) Simulated change in capacitance upon ligand adsorption to the intracellular membrane surface. (F) Representative capacitance change in SERT-expressing cells in response to 30 µM desipramine at pH 5.5 (left panel) and at pH 7.4 (right panel). (G) Comparison of $\Delta C_M$ upon application of 30 µM desipramine at pH 7.4 and pH 5.5. The lines connect data recorded in the same cell: pH 7.4: 111 ± 122 fF; pH 5.5: −496 ± 179 fF; n = 6; p=0.03, Wilcoxon test. (H) Representative capacitance change in response to 30 µM desipramine at pH 5.5 (left panel) and at pH 7.4 (right panel) in a control cell. Paired measurements in seven control cells showed no change upon application of 30 µM desipramine at pH 5.5, and a change of 332 ± 68 fF at pH 7.4 (data not shown).

DOI: https://doi.org/10.7554/eLife.34944.015

The following figure supplement is available for figure 7:

**Figure supplement 1.** Cocaine-induced displacement currents recorded from HEK293 cells expressing the dopamine transporter.

DOI: https://doi.org/10.7554/eLife.34944.016

presence of intracellular desipramine. This was the case: at pH 5.5, 30 µM desipramine caused a decrease in apparent $C_M$ (left panel in *Figure 7F*). In contrast, at pH 7.4, 30 µM desipramine elicited an increase in capacitance (right panel in *Figure 7F*). It is also evident from this representative trace (right panel in *Figure 7F*) that the increase at pH 7.4 was preceded by an initial decrease. This can be explained as follows: the initial decrease is the consequence of desipramine binding to SERT; the subsequent increase results from accumulation of intracellular desipramine. When assessed in paired recordings at pH 7.4 and pH 5.5, 30 µM desipramine consistently produced a drop in capacitance at pH 5.5 but not at pH 7.4 (*Figure 7G*). We also repeated these experiments in control cells (*Figure 7H*). As expected, desipramine increased the capacitance at pH 7.4 ($\Delta C_M$ ~300 fC) but was ineffective at pH 5.5.

## Ibogaine binds to an extracellular site on SERT

Most inhibitors of SERT bind to the outward-facing conformation. However, ibogaine is one notable exception. Ibogaine stabilizes the inward-facing conformation of SERT and inhibits substrate uptake in a non-competitive fashion (*Jacobs et al., 2007*). Ibogaine was therefore assumed to act on SERT by binding to a site in the inner vestibule. This implies that ibogaine gains access to the inner vestibule through the cytosol. We examined this hypothesis by measuring the effect of ibogaine on $C_M$ (*Figure 8*): application of 10 µM ibogaine resulted in a reduction of the $C_M$ (upper panel in *Figure 8A*). This effect was absent in control cells (lower panel in *Figure 8A*). Hence, these observations unequivocally refute the idea that ibogaine binds to a cytosolic site on SERT. Instead, our data suggest that the ibogaine binding site is directly accessible from the extracellular face of the transporter. We also evaluated the ibogaine-induced change in $C_M$ in paired recordings at pH 5.5 and 7.4, which did not affect the response to ibogaine (*Figure 8B*), although the difference in pH is expected to substantially decrease the fraction of non-protonated, membrane permeable ibogaine (pKa = 8.05).

## Discussion

The Gouy-Chapman model was developed more than a century ago to understand the distribution of diffusible ions residing on top of a charged and polarizable surface (*Gouy, 1909*; *Chapman, 1913*). While originally developed for a description of the interaction between dissolved ions and a metal electrode, this model also provides a conceptual framework to approach the electrical properties of a lipid bilayer. Fixed charges arise primarily from the net negative charge of phospholipid head groups (*McLaughlin et al., 1971*). In addition, proteins which reside in or at the plasma

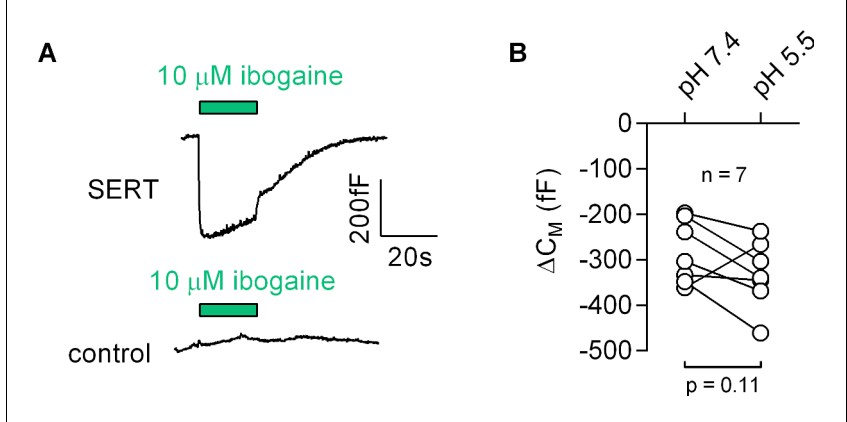

**Figure 8.** Ibogaine binds to an extracellular site of SERT. (A) Representative capacitance change induced by 10 µM ibogaine in a SERT-expressing cell (upper panel) and a recording in a control cell (lower panel). (B) A comparison of the ibogaine-induced reduction in $C_M$ at pH 7.2 and 5.5. The lines connect data points measured in the same cell (n = 7). The capacitance changes were 283 ± 70 fF and 331 ± 73 fF at pH 7.4 and pH 5.5, respectively (means ±S.D.). These values were not significantly different (p=0.11; Wilcoxon test).
DOI: https://doi.org/10.7554/eLife.34944.017

membrane, also contribute to the surface charges (*Green and Andersen, 1991*; *Madeja, 2000*). In cells, these surface charges are asymmetrically distributed with the inner membrane leaflet carrying a higher negative surface charge density. The electrostatic forces between the dissolved ions and the fixed surface charges of the membrane drive the formation of a Gouy-Chapman diffusive layer. Thus, the underlying theory predicts, under voltage-clamp conditions, any binding reaction which results in adsorption of a charged ligand to the surface of the cell membrane must evoke a displacement current and an apparent change in capacitance. In spite of the veneration of the Gouy-Chapman model, the current experiments are – to the best of our knowledge - the first to demonstrate that binding reactions to surface proteins can be directly monitored by recording the predicted currents and the apparent changes of the membrane capacitance.

In the present study, we exploited the rich pharmacology of SERT to test the predictions of the Gouy-Chapman model: (i) Inhibitors such as cocaine and desipramine produced a displacement current and the related change in apparent $C_M$. Their binding per se did not result in movement of mobile charges within the membrane electric field and was fully accounted for by the change in outer surface charge density caused by ligand adsorption. This conclusion is supported by the negative slope of the current-voltage relation. In addition, the change in $C_M$ is consistent with the average density of SERT in the cell line employed: adsorption of one charged ligand/molecule gives rise to a change in $C_M$ of about 500 fF. (ii) Displacement currents were also seen with the substrate serotonin provided that chloride was omitted to eliminate the conformational change associated with substrate translocation. (iii) Finally, we used ibogaine to interrogate the Gouy-Chapman model; binding of ibogaine also produced a drop in apparent $C_M$. While inhibitors bind to the outward-facing conformation, ibogaine traps SERT in the inward-facing conformation (*Jacobs et al., 2007*; *Bulling et al., 2012*). The precise binding site of ibogaine is unknown, but our capacitance recordings unequivocally refute the original hypothesis that ibogaine gains access to the inner vestibule. Our findings are also consistent with earlier observations, which showed that ibogaine was ineffective when administered to SERT via the patch electrode (*Bulling et al., 2012*).

SERT ligands are protonated amines, which form an ion pair with D98 in the orthosteric S1 binding site (*Coleman et al., 2016*). Because of the large number of transporters expressed on the surface of our HEK293 cell line, adsorption of one ligand per transporter suffices to account for our experimental findings, both qualitatively and quantitatively. However, it is plausible that the binding of an uncharged ligand can also be detected by a capacitance change provided that ligand binding results in conformational rearrangements, which alter the solvent accessible surface of the protein. We note that on average the changes in apparent capacitance were somewhat larger with cocaine (495 fF, n = 29) than with ibogaine (283 fF, n = 7; p=0.0007, Mann-Whitney U-test). This difference presumably reflects the difference in accessible surface charges in the cocaine- and the ibogaine-bound states. Finally, it is worth pointing out that capacitance measurements are more specific for monitoring binding events than the recording of the displacement current. The displacement currents evoked by cocaine, desipramine and 5-HT (in the absence of Cl$^-$) validate the predictions of the Gouy-Chapman model. However, the small size and transient occurrence of these currents limit their applicability for the assessment of ligand/protein interactions. Measurement of the apparent change in $C_M$ is a better technique to investigate ligand binding because of the sustained amplitude change (*Figure 5*). An additional advantage of this readout is that measurements of $C_M$ are less susceptible to the masking effects of electrogenic transitions associated with conformational change. This was also seen in our recordings: the 5-HT-induced peak current was approximately three to four times larger than the cocaine peak, because most of the current elicited by 5-HT was produced by charges moving in response to conformational change, despite the small associated valence (0.15; *Hasenhuetl et al., 2016*). In contrast, the extent by which the capacitance was reduced was similar for cocaine and 5-HT (*Figure 5F*). This can be explained as follows: the small valence associated with the mobile charge in SERT means that only a small fraction of this charge can be moved by the 80 mV voltage step used in our protocol to measure membrane capacitance. In fact, the simulation summarized in *Figure 5—figure supplement 2* documents that the current is too small to affect the analysis, which allows for extraction of total capacitance and changes therein. We stress that measurements of capacitance rather than of ligand-induced transient currents allow for obtaining the true binding affinity. In this context, we also want to note that we estimated the minimal expression density of the target membrane protein required for detection of the ligand-induced change in apparent $C_M$ to be 20,000 units/pF (see Materials and methods).

We stress that adsorption of a ligand to a surface protein does not evoke an actual change in $C_M$. Accordingly, we refer to the recorded alterations as apparent changes in capacitance. The point is illustrated by the equivalent circuit of a cell (*Figure 9*): the schematic representation depicts a battery ($V\phi$), which is connected in series with a capacitor ($C_M$) representing the cell membrane. This battery was included to account for the voltage drop caused by the asymmetry of the surface charge densities at the outer and inner membrane leaflets. Importantly, this battery is the element in the circuit, which is affected by charged ligand adsorption. The physiological relevance of this battery in a living organism is readily appreciated: it is, for instance, known that the voltage for half-maximal activation ($V_{0.5}$) of voltage-gated ion channels can be shifted by the presence of divalent ions in the extracellular fluid (*Frankenhaeuser and Hodgkin, 1957*). This is the reason why hypocalcemia causes seizures. The shift in $V_{0.5}$ is accounted for by the screening of surface charges by dissolved ions, which - similar to charged ligand adsorption - impinges on the highlighted battery. However, while surface charge screening is a long-range effect of ions that keep their hydration shell, ligand adsorption reflects a more specific interaction: it relies on tight binding of a ligand, which requires partial shedding of its water shell. We stress that in the case of surface charge screening no reference is made to the concept of binding affinities, which - in contrast - is implicit to charged ligand-adsorption to surface proteins.

To study the kinetics of ligand binding, capacitance recordings are not *a priori* limited to charged ligands. In fact, any reaction which alters the surface charge in the membrane can be detected by recording the apparent change in capacitance. This conclusion is also supported by our observations that desipramine increased the apparent $C_M$ at concentrations exceeding 10 µM. Three lines of evidence indicate that this was a non-specific action: (i) the increase in capacitance was seen regardless of whether cells expressed SERT or not; (ii) it was not saturable and (iii) it was not mimicked by the other SERT ligands (cocaine, serotonin and ibogaine). The most plausible explanation is to assume enrichment of desipramine at the inner leaflet of the membrane. The polar surface area is a good predictor of membrane permeability (*Palm et al., 1998*). The polar surface areas for desipramine, cocaine and 5-HT are 15 $Å^2$, 56 $Å^2$ and 62 $Å^2$, respectively (values taken from https://www.ncbi.nlm.nih.gov/pccompound). Thus, the low polar surface area of desipramine is consistent with its rapid permeation and the resulting change in capacitance. This is further supported by the findings of another study (*Sheetz and Singer, 1974*), which showed that amphiphilic cations, such as desipramine, accumulate at the inner membrane surface given that the surface charge density of the plasma membrane is more negative in the internal leaflet.

Incidentally, these observations highlight applications of possible $C_M$ recordings, which go beyond measuring ligand binding in real time: there are many biological processes known to alter the amount and the distribution of solvent accessible charge in the membrane. This (non-exhaustive) list includes: (i) reactions which are orchestrated by flippases and floppases and maintain the vital asymmetry between inner and outer surface-charge densities (*Zachowski et al., 1989*; *Groen et al., 2011*); (ii) dissipation of an existing charge-asymmetry by scramblases (*Zachowski, 1993*; *Suzuki et al., 2010*; *Suzuki et al., 2013*); (iii) alteration of the charge density at the inner surface by enzymes, e.g. phospholipase C via cleavage of $PIP_2$ (*Mauco et al., 1979*; *Rittenhouse-Simmons, 1979*), lipid kinases (*Pike and Arndt, 1988*) and phosphatases (i.e. $P_{ten}$, *Maehama and Dixon, 1998*). These activities ought to be amenable to recordings of the membrane capacitance given that they result in a sufficiently large change in surface charge density. Thus, capacitance measurements may be

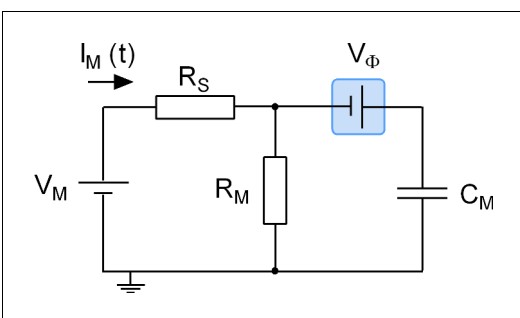

**Figure 9.** Equivalent circuit of the cell. The battery denoted $V_M$ accounts for the difference in potential between the intra- and extracellular bulk solutions. $R_S$ is the access resistance of the patch electrode, $R_M$ is the electrical resistance of the membrane and $C_M$ is the electrical capacitance thereof. Highlighted in blue is a second battery denoted by $V_\Phi$. This battery accounts for the potential difference created by the asymmetry in the intra- and extracellular surface charge densities and is the element in the circuit, which is affected by charged ligand adsorption.
DOI: https://doi.org/10.7554/eLife.34944.018

useful to extract kinetic information on these key biological reactions in real time and by a label-free approach.

## Materials and methods

### Whole-cell patch-clamp recordings

Recordings of tetracycline-inducible HEK293 cells stably expressing a GFP-tagged human serotonin transporter (hSERT) were performed. HEK293 cells have been previously authenticated by STR profiling at the Medical University of Graz (Cell Culture Core Facility). The cells were regularly tested for mycoplasma contamination by DAPI staining. The cells were maintained in Dulbecco's Modified Eagle's Medium (DMEM) containing 10% fetal bovine serum and selection antibiotics (150 $\mu g \cdot ml^{-1}$ zeocin and 6 $\mu g \cdot ml^{-1}$ blasticidin). Twenty-four h prior to the experiment, the cells were seeded onto poly-D-lysine-coated dishes (35 mm Nunc Cell-culture dishes, Thermoscientific, USA) containing medium supplemented with 1 $\mu g \cdot ml^{-1}$ tetracycline. If not stated otherwise, the cells were continuously superfused with an external solution containing 140 mM NaCl, 3 mM KCl, 2.5 mM $CaCl_2$, 2 mM $MgCl_2$, 20 mM glucose, and 10 mM HEPES (pH adjusted to 7.4 with NaOH). In some instances the following changes were made: (i) for experiments at pH 5.5, HEPES was substituted by 2-(N-morpholino)ethansulfonic-acid; (ii) for experiments in $Na^+$-free external solution, NaCl was replaced by $NMDG^+$(N-methyl-D-glucamine)$Cl^-$; (iii) for experiments in $Cl^-$-free external solution, NaCl was replaced by $Na^+MES^-$ (methanesulfonate). The internal solution in the patch pipette contained 152 mM NaCl, 1 mM $CaCl_2$, 0.7 mM $MgCl_2$, 10 mM HEPES, and 10 mM EGTA (ethylenglycol-bis(aminoethylether)-N,N,N′,N′-tetra-acidic-acid) (pH 7.2 adjusted with NaOH). Where indicated, the internal $Cl^-$ concentration was reduced (152 mM NaOH, 1 mM $CaCl_2$, 0.7 mM $MgCl_2$, 10 mM EGTA, 10 mM HEPES, pH 7.2 adjusted with methaneosulfonic acid). Ligands were applied via a four-tube or eight-tube ALA perfusion manifold using the Octaflow perfusion system (ALA Scientific Instruments, USA), which allowed for complete solution exchange around the cells within 100 ms. Currents were recorded at room temperature (20–24°C) using an Axopatch 200B amplifier and pClamp 10.2 software (MDS Analytical Technologies). Current traces were filtered at 1 kHz and digitized at 10 kHz using a Digidata 1440 (MDS Analytical Technologies). The recordings were analyzed with Clampfit 10.2 software. Passive holding currents were subtracted, and the traces were additionally filtered using a 100 Hz digital Gaussian low-pass filter.

### Membrane capacitance measurements

Recordings were performed in the whole-cell configuration. For the measurement we used a train of square wave voltage pulses with an amplitude ±40 mV and a frequency of 200 Hz. Bipolar square waves were used for measuring capacitance, because they are less sensitive to changes in series resistance ($R_S$) and membrane resistance ($R_M$) when compared with other methods for high-resolution membrane capacitance recording (*Lindau and Neher, 1988*; *Thompson et al., 2001*). In the protocol the holding potential was set to 0 mV if not stated otherwise. Exponential current responses were low pass filtered by a 10 kHz Bessel filter and sampled at 100 kHz rate. The cross talk between $R_S$ and $C_M$ was further suppressed as follows: the acquired current traces were first deconvoluted with the transfer function of the recording apparatus and the passive membrane parameters of a cell were calculated from the theoretical function as described elsewhere (*Hoťka and Zahradník, 2017*). This method provides reliable $C_M$ estimates even in conditions where large changes in $R_S$ occur (1 MΩ change in $R_S$ produces a $C_M$ change smaller than 1 fF; *Hoťka and Zahradník, 2017*). The pipette capacitance was recorded in the cell-attached mode first and subtracted from the currents recorded in the whole-cell configuration prior to further analysis. The patch pipettes used had a resistance of 2–4 MΩ. To stabilize the level of stray capacitance, the pipettes were coated with hydrophobic resin Sylgard184 (Dow Corning, USA). The method of bipolar square wave stimulation can be rendered largely insensitive to changes in stray capacitance ($C_S$) by ignoring the current during the time when the stray capacitor charges (*Lindau and Neher, 1988*). It takes 60 μs to fully charge the stray capacitor, which only affects the first six sample points of the recorded current (sampling rate = 100 kHz). In contrast, the time required to charge the cell membrane is in the order of hundreds of microseconds. By removing the first six data points it is therefore possible to provide estimates of $C_M$, which are insensitive to changes in $C_S$ (*Thompson et al.,*

*2001*; *Novák and Zahradník, 2006*; *Hoťka and Zahradník, 2017*). A detailed description of the method can be found in *Burtscher et al. (2019)*.

## Modelling

The membrane capacitance of SERT-expressing HEK293 cells was estimated using bipolar square wave stimulation. The voltage step of amplitude $V_M$ drives the translocation of capacitive charge. However, according to the Gouy-Chapman theory, the transmembrane potential ($\Phi_t$) is not equal to $V_M$ (*Figure 4—figure supplement 1*). $\Phi_t$ deviates from $V_M$, by $(\Phi_i - V_M - \Phi_o)$. Where $\Phi_i$ and $\Phi_o$ are the potentials at the inner and outer surface, respectively.

The potentials on each side of a membrane ($\Phi_i$ and $\Phi_o$)) are coupled by capacitive charges. To calculate these potentials, we adopted the equations from Genet and co-workers (*Genet et al., 2000*). These coupled equations describe the relation between the mobile (Q) and fixed ($\sigma$) membrane charge densities and the potential drops of the bulk regions ($\Phi$). We used the notation according to Plaksin and co-workers (*Plaksin et al., 2017a*, *Plaksin et al., 2017b*).

$$\left(\sigma_o - \frac{\varepsilon_b}{\delta_b}(\Phi_i - \Phi_o)\right)^2 = 2\varepsilon_{Sol}RT\sum_{j=1}^{n} c_j^o(\infty) \times \left(e^{\left(\frac{z_j^o F \Phi_o}{RT}\right)} - 1\right) \tag{1}$$

$$\left(\sigma_i + \frac{\varepsilon_b}{\delta_b}(\Phi_i - \Phi_o)\right)^2 = 2\varepsilon_{Sol}RT\sum_{j=1}^{n} c_j^i(\infty) \times \left(e^{\left(-\frac{z_j^i F}{RT}(\Phi_i - V_M)\right)} - 1\right) \tag{2}$$

Where $\varepsilon_{Sol}$ is a dielectric constant of a bulk solution; $R$ and $F$ are the ideal gas and Faraday constants; T is the absolute temperature; $c_j(\infty)$ are the intracellular and extracellular ion concentrations far from the membrane and $z_j$ is ion valence. The equations were solved numerically for $\Phi_i$ and $\Phi_o$. These values were used to calculate the transmembrane voltage: $\Phi_t = (\Phi_i - \Phi_o)$.

The cocaine binding to a SERT-expressing HEK293 cell was modeled as the reduction of outer surface charge density proportional to the number of transporters with a single binding site for cocaine (representing 1 surface charge). This reduction in a surface charge density gives rise to a change in $\Phi_o$ and via Eq. 1 and Eq. 2 also in $\Phi_i$.

The experimentally observed ligand-induced currents were modeled as capacitive currents resulting from the change of $\Phi_t$ caused by ligand application. Their amplitudes are proportional to the total membrane capacitance of a cell $C_{theoretical}$ and the amplitude of an induced voltage change:

$$i(t) = C_{theoretical} \frac{d(\Phi_i - \Phi_o)}{dt} \tag{3}$$

The same model was also used to simulate the observed changes in membrane capacitance.
The calculated voltage $\Phi_t = (\Phi_i - \Phi_o)$ is used to estimate the total capacitive charge:

$$Q = \frac{\Phi_t A \varepsilon_b}{\delta_b} \tag{4}$$

In the real recording situation, the estimation of $C_M$ from exponential current responses elicited by voltage steps assumes that the applied voltage $V_M$ is equal to the voltage drop across the membrane $\Phi_t$. To mimic the real measurement, we keep this assumption and we calculate the $C_M$ estimates from the total charge Q using a theoretical value of $V_M$ by equation:

$$C_M = Q/V_M \tag{5}$$

Upon ligand application, surface charge density on the outer leaflet is reduced. Therefore, the transmembrane voltage $\Phi_t$ Eq. 5 is altered. The use of $V_M$ Eq. 5 instead of $\Phi_t$ Eq. 5 in *Eq. 5* thus gives rise to apparent changes in membrane capacitance.

## Sensitivity analysis

The average standard deviation of the capacitance signal was derived from 12 independent measurements. Each trace analyzed had a time length of 5 s (sampling rate = 50 Hz). The average SD calculated was 4.84 ± 1.16 fF. We assumed a detectable change in capacitance amplitude if this

exceeded two SDs. We used this value to estimate the minimal expression density of the target membrane protein required for detection of ligand-induced change in apparent membrane capacitance. The calculated density amounted to approximately 20,000 units/pF. However, this value applies only if the ligand neutralizes one surface charge on the protein. Accordingly, the lowest expression density required to detect change in apparent capacitance is decreased if a charged ligand adsorbs to more surface charges on the target protein.

### Modeled and corrected apparent association rates

The apparent association rate ($k_{app}$) of cocaine to SERT was calculated based on the previously estimated association and dissociation rate constants (*Hasenhuetl et al., 2015*) and was plotted as a function of the cocaine concentration (dashed line in *Figure 4D*). The rates were modeled as: $k_{app} = k_{on}*[Cocaine] + k_{off}$. We attribute the discrepancy between the theoretical and the measured rates to the finite solution exchange rate of our application device ($\sim 20$ s$^{-1}$). To model the corrected rate, we used a two-state model and simulated pseudo-first order binding assuming an exponential rise in cocaine concentration with the rate of solution exchange.

## Acknowledgements

We thank Shreyas Bhat, Klaus Schicker, and Peter S Hasenhuetl for discussions and comments on the data. This work was supported by the Austrian Science Fund/FWF Project P28090 (to W S), Project Program Grant SFB35 (F3510 to M F). The authors declare that they have no conflicts of interest with the contents of this article.

## Additional information

### Funding

| Funder | Grant reference number | Author |
|---|---|---|
| Austrian Science Fund | P28090 | Walter Sandtner |
| Austrian Science Fund | F3510 | Michael Freissmuth |

The funders had no role in study design, data collection and interpretation, or the decision to submit the work for publication.

### Author contributions

Verena Burtscher, Conceptualization, Data curation, Formal analysis, Writing—original draft; Matej Hotka, Conceptualization, Formal analysis, Methodology, Writing—original draft; Yang Li, Data curation; Michael Freissmuth, Conceptualization, Funding acquisition, Writing—original draft; Walter Sandtner, Conceptualization, Formal analysis, Supervision, Funding acquisition, Writing—original draft

### Author ORCIDs

Walter Sandtner  http://orcid.org/0000-0003-3637-260X

### Decision letter and Author response

Decision letter https://doi.org/10.7554/eLife.34944.022
Author response https://doi.org/10.7554/eLife.34944.023

## Additional files

### Supplementary files

• Supplementary file 1. Table of model parameters.
DOI: https://doi.org/10.7554/eLife.34944.019

• Transparent reporting form
DOI: https://doi.org/10.7554/eLife.34944.020

## Data availability

All data generated or analysed during this study are included in the manuscript and supporting files. Source data files have been provided for Figures 1 and 5.

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
