## [Decision Letter]

Thank you for submitting your article "A label-free approach to detect ligand binding to cell surface proteins in real time" for consideration by *eLife*. Your article has been favorably evaluated by Richard Aldrich (Senior Editor) and three reviewers, one of whom, Baron Chanda (Reviewer #1), is a member of our Board of Reviewing Editors. The following individuals involved in review of your submission have agreed to reveal their identity: Andrew J R Plested (Reviewer #2); Leon D. Islas (Reviewer #3).

The reviewers have discussed the reviews with one another and the Reviewing Editor has drafted this decision to help you prepare a revised submission.

Summary:

This is an interesting manuscript that describes a new method to probe direct binding of ligands to cell surface proteins. In many cases, ligands or modulators of protein function are charged molecules which when they bind will change the surface charge density either by neutralizing existing charges or by adding more charge to the membrane. This will be manifested as a change in membrane capacitance which typically has not been directly observed largely due to the fact that the membrane proteins are not expressed at sufficient levels. In this study, Burtscher et al. show that the binding of cocaine and other molecules to serotonin (5-HT) transporter SERT result in a capacitive current which is not due to displacement of charges but rather due to modification of surface charges. While the reviewers agree that the properties of the inward currents recorded here are entirely consistent with the predictions of the simple Guy-Chapman model of the double layer, they have raised few concerns which will have to be addressed in the revised version.

Essential revisions:

1) The negative slope of the current-voltage curve is intriguing and is the critical piece of evidence to support the claim that the increased capacitance is due to modification of surface potential. The increases are small but more importantly, without showing complete traces, it is not possible to determine where the baseline is. For instance, it is not clear whether the linear component of membrane capacitance was properly compensated or is some of it an artifact of compensation.

2) Related to the above point – the authors measure cell capacitance by estimating the charging of the membrane RC circuit due to square voltage pulses. This method has to rely on a good estimate of the leak current and is very sensitive to series resistance (R_s_) compensation. The authors should demonstrate that R_s_ is not drifting during the recordings. Also, measuring capacitance in this fashion is affected greatly by even subtle changes in the level of the solution, which is a problem in experiments that use perfusion of solutions. This problem is present even when using Sylgard-coated pipettes. Please make a better presentation and discussion of the problems of estimating capacitance as done in these experiments.

3) Although the concentration of 5-HT and cocaine used are small, given the large expression level attained, it is important to estimate what fraction of the transient inward currents are caused by actual transport of these substrates. Is it possible that some fraction of the charge is actual current carried by 5-HT and cocaine, which should be positively charged?

4) In the data of Figure 7, the capacitance is different at 0, 50 or -50 mV, even in untransfected cells, please explain the possible origin of this.

5) The modeling and theoretical treatment is a major plus – although it's not well enough explained in my view. For example, "this idea was incorporated into the model" – how? There is no explanation how the simulations were done, that I found. Also, in the subsection “The voltage dependence of the cocaine-induced capacitance change is predicted by the Gouy-Chapman model” – It would be good to have some quantification of how the model performs. What are the parameters?

6) The first four figures are rather simple and might be condensed into two – in order to get faster to the real meat of the argument. It should be emphasized that capacitance measurement is more robust than current measurement and it is possible to obtain the true binding affinity.

---

## [Author Response]

Essential revisions:1) The negative slope of the current-voltage curve is intriguing and is the critical piece of evidence to support the claim that the increased capacitance is due to modification of surface potential. The increases are small but more importantly, without showing complete traces, it is not possible to determine where the baseline is. For instance, it is not clear whether the linear component of membrane capacitance was properly compensated or is some of it an artifact of compensation.

We would like to stress that the measured currents were induced by rapid application of a ligand to cells expressing SERT and not by a step change in voltage. It is only necessary to subtract (compensate for) “passive” displacement currents in the latter case. Additionally and in response to the reviewer's criticism, we show larger segments of the original traces to clearly visualize the baseline in the new Figure 3C and 3E (original Figure 4C and 4E).

2) Related to the above point – the authors measure cell capacitance by estimating the charging of the membrane RC circuit due to square voltage pulses. This method has to rely on a good estimate of the leak current and is very sensitive to series resistance (R_s_) compensation. The authors should demonstrate that R_s_ is not drifting during the recordings. Also, measuring capacitance in this fashion is affected greatly by even subtle changes in the level of the solution, which is a problem in experiments that use perfusion of solutions. This problem is present even when using Sylgard-coated pipettes. Please make a better presentation and discussion of the problems of estimating capacitance as done in these experiments.

We show the time dependent evolution of C_M_, R_M_ and R_S_ in a representative recording (see Figure 5B); the traces allow the reader to appreciate how the circuit parameters in our measurements change over time. It is evident that there is no cross talk between these parameters.

We explicitly stated this as follows:

“Figure 5B shows a representative recording of the membrane capacitance (C_M_) with the two other circuit parameters R_M_ and R_S_ upon application and subsequent removal of 100 µM cocaine to HEK293 cells expressing SERT. It is evident from this recording that there was no cross talk between circuit parameters.”

In addition, we justify the choice of the bipolar square-wave method for measuring capacitance by a more detailed description; the pertinent text now reads:

"Bipolar square waves were used for measuring capacitance, because they are less sensitive to changes in R_S_ and R_M_ when compared to other methods for high-resolution membrane capacitance recording (Lindau and Neher, 1988, Thompson et al., 2001). […] This method provides reliable C_M_ estimates even in conditions where large changes in R_S_ occur (1MΩ change in R_S_ produces a C_M_ change smaller than 1 fF; see Hotka and Zahradník, 2017)."

In addition, we also explicitly address the concern of stray capacitance by the following explanation:

"The method of bipolar square-wave stimulation can be rendered largely insensitive to changes in stray capacitance (C_S_) by ignoring the current during the time when the stray capacitor charges (Lindau and Neher, 1988). […] By removing the first six data points it is therefore possible to provide estimates of C_M_, which are insensitive to changes in C_S_ (Thompson et al., 2001, Novak and Zahradnik, 2006, Hotka and Zahradnik 2017)."

3) Although the concentration of 5-HT and cocaine used are small, given the large expression level attained, it is important to estimate what fraction of the transient inward currents are caused by actual transport of these substrates. Is it possible that some fraction of the charge is actual current carried by 5-HT and cocaine, which should be positively charged?

We stress that only 5-HT is a substrate of SERT. In contrast, cocaine is an inhibitor, which is not transported. We explicitly state this:

"In contrast to 5-HT, which is the cognate substrate of SERT, cocaine is not translocated. Under our recording conditions, both compounds give rise to a transient current, which may arise by two alternative mechanisms …"

We hope that this addresses the question of the reviewer in an adequate manner.

4) In the data of Figure 7, the capacitance is different at 0, 50 or -50 mV, even in untransfected cells, please explain the possible origin of this.

In the main manuscript, we explicitly alert the reader to the voltage-dependent change in capacitance by stating:

"It is important to note that the Gouy-Chapman model also predicted voltage-dependent change in capacitance observed in control cells (Figure 6—figure supplement 1)."

We added a figure (Figure 6—figure supplement 1) showing the dependence of the membrane capacitance on V_M_ as predicted by the Gouy-Chapman model (Materials and methods).

In the legend of Figure 6—figure supplement 1, we address the reviewer’s question as follows:

"The cell membrane capacitance is composed of a voltage-independent component that reflects the lipid bilayer and a voltage-dependent component that originates primarily from voltage-dependent proteins in the plasma membrane (e.g. voltage-gated ion channels). […] This is evident from an inspection of the equation of the Gouy-Chapman model (see Materials and methods).”

5) The modeling and theoretical treatment is a major plus – although it's not well enough explained in my view. For example, "this idea was incorporated into the model" – how? There is no explanation how the simulations were done, that I found. Also, in the subsection “The voltage dependence of the cocaine-induced capacitance change is predicted by the Gouy-Chapman model” – It would be good to have some quantification of how the model performs. What are the parameters?

In the manuscript we modeled two phenomena: (i) ligand-induced transient currents and (ii) ligand-induced apparent changes in membrane capacitance. We now added Figure 4—figure supplement 1 and Figure 5—figure supplement 1 to illustrate how we modeled the transient current and apparent capacitance changes, respectively.

The corresponding figure legends provide a detailed description:

Figure 4—figure supplement 1 legend:

“Schematic representation of the voltage across the membrane as predicted from the Gouy-Chapman model.[…] The parameters used for the model are provided in Supplementary file 1.”

Figure 5—figure supplement 1 legend:

“Modeling of the ligand-induced change in membrane capacitance. The rectangular square pulses give rise to a current, which allows for estimating membrane capacitance by extracting the difference (V_step_) between the transmembrane voltages at the two holding voltages (here -40 and +40 mV). […] The bottom panels illustrate how the cocaine-induced increase in positive charges on the outer surface translates into the change in capacitance. The parameters used are provided in Supplementary file 1.”

The model parameters used are available in Supplementary file 1.

Additionally, we incorporated into the Materials and methods section the description of the Gouy-Chapman Model:

“Modelling.The membrane capacitance of SERT expressing HEK293 cells was estimated using bipolar square wave stimulation. […] The use of V_M_ instead of Ф*_t_* in the Equation 5 thus gives rise to apparent changes in membrane capacitance. “

6) The first four figures are rather simple and might be condensed into two – in order to get faster to the real meat of the argument. It should be emphasized that capacitance measurement is more robust than current measurement and it is possible to obtain the true binding affinity.

We merged the Figures 1 and 2 as suggested. As recommended, we also emphasize in the text that capacitance measurements are more robust than current measurements and that it is possible to obtain the true binding affinity. The pertinent sentence reads:

“We stress that measurements of capacitance rather than ligand-induced transient currents allow for obtaining the true binding affinity.”